

# Agility-based exercise training compared to traditional strength and balance training in older adults: a pilot randomized trial

Eric Lichtenstein[1], Mareike Morat[2], Ralf Roth[1], Lars Donath[2] and Oliver Faude[1]

[1] Department of Sport, Exercise and Health, University of Basel, Basel, Switzerland
[2] Department of Intervention Research in Exercise Training, Institute of Training and Computer Science in Sport, German Sport University Cologne, Cologne, Germany

Corresponding author
Eric Lichtenstein,
e.lichtenstein@unibas.ch

## ABSTRACT

**Background**. In addition to generally high levels of physical activity, multi-component exercise training is recommended for the maintenance of health and fitness in older adults, including the prevention of falls and frailty. This training often encompasses serial sequencing of balance, strength, endurance and other types of exercise. Exercise training featuring integrative training of these components (i.e. agility training) has been proposed, as it more likely reflects real life challenges like stop-and-go patterns, cutting manoeuvers, turns and decision-making. In this study, we compared the efficacy of an agility-based training to the traditional strength and balance training approach with regard to selected risk factors for falls and frailty.

**Methods**. We trained twenty-seven community-dwelling healthy seniors (16♂; 11♀; age: 69.5 ± 5.3 y; BMI: 26.4 ± 3.7 kg/m$^2$) for 8 weeks in a group setting with 3 sessions per week, each lasting 50 minutes. Participants were randomized into either the agility group (AGI; $n = 12$), that used the integrative multi-component training, or the traditional strength and balance group (TSB; $n = 15$). TSB performed balance and strength exercises separately, albeit within the same session. The training of both groups progressively increased in difficulty. Outcomes were static and dynamic balance (single leg eyes open stand, Y-balance test, reactive balance), lower limb (plantar flexion and dorsal extension) and trunk flexion and extension maximum strength and rate of torque development (RTD). In addition, we tested endurance by the six-minute walk test (6MWT). We calculated linear mixed effects models for between-groups comparisons as well as effect sizes (ES) with 95 % confidence intervals.

**Results**. Small ES in favor of AGI were found for plantar flexion strength (ES > 0.18[−0.27;0.89]) and RTD (ES > 0.43[−0.19;1.36]) as well as trunk extension RTD (ES = 0.35[−0.05;0.75]). No other parameters showed notable between group differences. Compliance was high in both groups (AGI: 90 ± 8% of sessions; TSB: 91 ± 7% of sessions).

**Discussion**. Agility-based exercise training seems at least as efficacious as traditional strength and balance training in affecting selected physical performance indicators among community-dwelling healthy seniors. In particular, lower limb and trunk extension explosive strength seem to benefit from the agility training.

## INTRODUCTION

Two billion people are expected to be 60 years of age or older by 2050 (*United Nations, 2011*). They are faced with age-related declines in neuromuscular, cardiovascular and cognitive capacity that result in frailty and loss of independence (*Mendonca et al., 2017*). Exercise-based interventions have shown to attenuate declines in physical performance and delay the onset of frailty and associated comorbidities (*Apostolo et al., 2018*; *Gray & Butler, 2017*; *McPhee et al., 2016*; *Tricco et al., 2017*) as well as treat various health conditions such as cardiovascular disease (*Perk et al., 2012*). In addition, exercise interventions have been shown to help in the prevention of falls, a common and potentially severe traumatic event in older age (*Sherrington et al., 2017*).

Physical activity guidelines for older people mirror these findings incorporating the training of endurance, balance, strength and flexibility (*Chodzko-Zajko et al., 2009*; *Elsawy & Higgins, 2010*; *Nelson et al., 2007*). Yet, physical activity participation rates are low amongst older adults (*McPhee et al., 2016*) rendering the recommendations and their feasibility questionable. In fact, following the guidelines around six exercise sessions have to be conducted weekly (two for each of strength, balance and flexibility) excluding the endurance training. The separate training of those factors has been criticized due to huge time demands and lack of specificity compared to real life challenges. Multi-modal exercise training concepts were proposed (*Donath, Van Dieën & Faude, 2016*). Within these multi-modal approaches, all physical abilities should be trained simultaneously instead of one after the other or in separate sessions and cognitive challenges should be incorporated. This could allow the exercise training to more closely reflect activities of daily living as well as fall threatening situations and, thus, prepare older adults to situations where all those factors might be required simultaneously.

*Donath, Van Dieën & Faude (2016)* proposed an agility framework for exercise testing and training. It considers complex functional tasks, including stop-and-go as well as cutting movements, changes of direction, perceptional challenges, decision making, reaction tasks and, thereby, stresses the need for continuous progression of difficulty and complexity of exercises. A test assessing the agility construct was developed and validity with regard to traditional fall risk factors was recently investigated (*Lichtenstein et al., 2019*). Associations were found between the fall risk factors gait speed, endurance, as well as selected strength and balance parameters and the measurement of agility.

The aim of the current study was to apply the agility framework to an exercise training intervention and compare the efficacy to a traditional exercise training intervention particularly targeting strength and balance in older adults.

## MATERIALS & METHODS

### General design

The study design was a two arm randomized pilot trial with an allocation ratio of 1:1. During eight weeks, participants received three training sessions per week for eight weeks (24 training sessions) of 50 min duration with sessions' content depending on group allocation. Allocation to the groups was done after the pre-test and participants were stratified by age, sex, BMI, and performance in the six-minute walking test. An independent researcher otherwise not working on the project performed the randomisation. The staff conducting the different assessments as well as leading the exercise classes were not masked to group allocation.

### Participants

In order to participate in the study, participants had to be between 60 and 80 years of age and had to be able to attend the training sessions in a local gym. All participants had to be retired, were required to live independently and not have taken part in a structured exercise program within the last six months. Exclusion criteria were any diagnosed cardiologic, neurologic or orthopaedic conditions. The physical activity readiness questionnaire (PAR-Q) was used to test participants' eligibility for participation. Study assistants handed out leaflets featuring the study information to local sports clubs, gyms, seniors clubs and contacted potential participants from the department's database. Interested older adults were initially interviewed to check for eligibility. The study protocol was approved by the local ethics committee (Ethics Committee of North-Western and Central Switzerland; approval number: 2017-07940) and complied with the Declaration of Helsinki. All participants signed an informed written consent prior to the start of the study after receiving all relevant study information.

### Intervention

Both groups trained three times per week in a group setting for 50 min over 8 weeks. We did not demand training participation, but we controlled for it in all training sessions by means of an attendance list. We defined compliance as percentage of attended training sessions.

Both groups' training sessions started with an unspecific 5-minute warm-up and ended with a 5-minute cool down. The main content of the training sessions differed as follows: The agility-training group (AGI) trained with a modular agility course posing different challenges for balance, strength, stop-and-go, change of direction, jumps, and rotation capabilities. Four different courses were designed that were each utilized for six training sessions. Each course had five to seven tasks which were increased in difficulty each training. Courses had a certain theme attached to them with the first course focusing on rotations around poles, the second on jumping, the third on stop-and-go tasks and the fourth on cutting manouvers. Increases in difficulty were achieved by combining several exercise modes within one exercise (e.g., balancing while performing a strength exercise triggered by an acoustic signal), adding more demanding cognitive challenges or introducing more difficult variants of an exercise. For example, in the first session two leg jumps with

controlled landings were performed, in the second session two leg jumps with controlled landings over an obstacle, and in the third session plyometric two leg jumps over obstacles. Participants usually worked in pairs with one person going through the agility course while the other person remained at the starting point. The latter person received an additional task for the waiting period.

The control group (TSB) trained with conventional body weight strength and balance exercises that were also progressively increased in difficulty. Training was organized in a circuit style with eight exercises per session, three strength exercises and five balance exercises. Strength exercises targeted the lower body and the trunk. Difficulty was increased by increasing the number of repetitions and the range of motion of the exercises. Balance exercises were made more difficult by adhering to common progression principles reducing the base of support, reducing sensory input, introducing unstable surfaces and adding additional tasks (*Donath et al., 2016b*). An outline of the two training regimes can be found in Table 1.

## Procedures

We assessed anthropometric data, static and dynamic balance performance, lower limb and trunk maximum strength and rate of torque development, and endurance in that order before and after the intervention period. All testing session were conducted according to established standard operational procedures at our laboratory. Testing was performed at an intra-individually similar time of day in each participant.

Some evidence suggests that postural sway assessed during quiet stance can serve as a predictor for future falls in community-dwelling elderly individuals (*Oliveira et al., 2018*) and therefore we selected it as an outcome. We assessed static balance on a Kistler force platform (KIS, Type 9286BA, Winterthur, Switzerland) recording data at 120 Hz. All participants stood barefoot on their dominant leg with eyes open for 30 s. Leg dominance was determined by the procedure established by *Coren (1993)*. Participants were instructed to (a) remain as stable as possible, (b) focus on a marker on the wall (distance: 1.5 m; height: 1.75 m), and (c) place the hands on the iliac crests. Three trials were executed with one minute of rest in between. The path length displacement of the center of pressure (CoP) was used as measure for static balance performance. The best performance (shortest path length) of the three trials was used for analysis. Coefficients of variation (CoV) of 8.6% can be expected in this sample based on unpublished data from a previous own study (*Lichtenstein et al., 2019*).

To test participants' ability to deal with external perturbations, they were tested three times on the Posturomed (Haider Bioswing, Pullenreuth, Germany) device with one minute rest in between. The Posturomed is a movable platform attached to a solid frame with two dampened pendulums on each corner. The platform is initially locked in a stable position 2.5 cm away from its neutral position in medio-lateral direction. Once participants are stable in this starting position, the lock is released unexpectedly and participants are challenged with reducing the oscillation of the platform. The participants stand with both feet shoulder width apart and open eyes in the starting position. Hands were placed on the iliac crest. Trial duration was 10 s starting with the release of the lock. An accelerometer

**Table 1** Main part of the exercise programs for agility and traditional strength and balance training group (excluding warm up and cooldown).

| | | Agility based Training | Traditional Strength and Balance Training |
|---|---|---|---|
| Session 1 to 5 | Exercises | Walking based: change of directions, cuts, obstacle crossing, bench balancing | Static balance exercises in double leg stance |
| | | | Squats, calf raises, supported split squats |
| | | Double leg line jumps, bench step ups | |
| | Duration | ~30 s per exercise, 2 rounds, 5 exercises | 30 s exercise, 30 s pause, 3 rounds, 8 exercises |
| | Variations | Elevated balancing, obstacle height, speed, plane of movement | Perturbations |
| Session 6 to 10 | Exercises | Walking based: change of directions, cuts, obstacle crossing, line balancing, swiss cross and combinations of these | Static balance exercises in step or tandem stance |
| | | | Squats, calf raises, side lunges, split squats, crunches |
| | | Unstable lunges, ball dribble | |
| | Duration | ~30 s per exercise, 2 rounds, 7 exercises | 50 s exercise, 30 s pause, 2 rounds, 8 exercises |
| | Variations | Colour coding of movement directions, all of the previous | Perturbations |
| | | | Strength exercises on slightly unstable surface |
| Session 11 to 15 | Exercises | Walking based: change of directions, cuts, rotations, obstacle crossing, line balancing, swiss cross and combinations of these | Static and dynamic balance exercises in step or tandem stance |
| | | Single leg line jumps, bench step ups, orientation games | Step ups, calf raises, side lunges, split squats |
| | Duration | ~30 s per exercise, 3 rounds, 7 exercises | 35 s exercise, 30 s pause, 3 rounds, 8 exercises |
| | Variations | Sound coding of movement tasks and directions, walking modes, all of the previous | Cognitive tasks, unstable surfaces, arm balance |
| | | | Strength exercises on slightly unstable surface |
| Session 16 to 20 | Exercises | Walking based: obstacle crawl and crossing, catch through the course, beam balancing, orientation games, hurdling | Balance exercises in step or tandem stance |
| | | Sprints, lunges, step ups, lateral double leg jumps | Planks, squats, single leg calf raises, side lunges, split squats, bulgarian split squats |
| | Duration | ~30 s per exercise, 3 rounds, 8 exercises | 40 s exercise, 20 s pause, 3 rounds, 8 exercises |
| | Variations | All of the previous | Cognitive tasks, perturbations, arm balance |
| Session 21 to 24 | Exercises | Walking based: Song with coded tasks on words, catch through the course, orientation and reaction games | Dynamic balance exercises in tandem or single leg stance |
| | | Lateral and forward single leg jumps, skipping on unstable surface, lunges, ball throw and catch exercises | Squats, single leg calf raises, step ups, bulgarian split squats |
| | Duration | ~30 s per exercise, 3 rounds, 8 exercises | 55 s exercise, 20 s pause, 3 rounds, 8 exercises |
| | Variations | Cognitive tasks, ball dribbling while performing the tasks, all of the previous | Unstable surfaces, perturbations, arm balance |

(MicroSwing 6, Haider Bioswing, Pullenreuth, Germany) was attached to the bottom of the platform and records the platform's acceleration from which the sway path was calculated. Again, the shortest sway path over the three trials was used for analysis. The accelerometer records data at 50 Hz. In our previous study (*Lichtenstein et al., 2019*), we observed an average CoV of 7.7% for this procedure (unpublished data).

The Y-balance test (Functional Movement Systems, Chatha, USA) was used to assess functional balance performance. Participants stood on the stance platform that has three pieces of PVC pipe attached in anterior, posteromedial, and posterolateral reach directions, giving a Y-shaped setup and were instructed to push a plastic box along the plastic pipes
as far as possible with one foot in the three directions while maintaining balance on the standing leg. Two familiarisation trials were conducted for both legs and each direction, followed by three assessment trials. Hands had to be placed at the iliac crest, plastic boxes were only allowed to be touched on the vertical surface and could not be kicked. The distance between the furthest reaching positions of the box from the centre was recorded. The composite score for each trial was calculated as the average reach for all three directions divided by leg length multiplied by 100 (*Lai et al., 2017*). Leg length was measured by the distance from the ground to the pubic bone during upright stance assessed by a bubble level. The best composite score of the three trials was used for analysis. Plisky and colleagues reported CoV of 5.9% for the composite score of the Y-balance test (*Plisky et al., 2009*) and a minimal detectable change of 0.05 for the composite score in older adults (*Sipe et al., 2019*). We found a similar CoV in our unpublished data of 5.1% (*Lichtenstein et al., 2019*).

Ankle and trunk maximum and explosive strength was measured with a series of isometric tasks on an isokinetic device (Isomed 2000, D&R Ferstl GmbH, Hemau, Germany). Dorsal extension and plantar flexion of the ankle was assessed by positioning the participants in a supine position with hip and knee angles in a neutral position (0°) and the ankle at 10° plantar flexion with arms crossed in front of the chest. Participants were strapped to the device so that only plantar flexion and dorsal extension were possible. Both legs were tested starting with the dominant one determined as mentioned before. The predominant strategy to maintain postural control up to moderately challenging tasks is the utilization of the ankle, also in seniors (*Donath et al., 2016a*). Therefore, measurement of ankle strength was chosen in favour of the knee.

Trunk strength was assessed on the Isomed trunk adapter with a hip angle of 85° and a knee angle of 45°. Participants were fixed at the chest, knees and hip and were instructed to pull with their hands on a handle that is placed on the clavicular bone. Lastly, trunk rotation strength was assessed on the trunk rotation adapter with hip and knees fixed at 90° with the hands loosely placed in their lap. Participants were instructed to push with their shoulder against a pad in the left and right direction. For each test three trials were conducted that were preceded by one familiarisation trial. Participants were instructed to push as fast and hard as possible in the indicated direction (*Maffiuletti et al., 2016*). Data were sampled at 200 Hz. Torque data was smoothed with a 6th order Butterworth filter with a cut-off frequency of 200 Hz. Maximum rate of torque development was calculated as the maximum rise of torque over a 150-millisecond window over the force-time relationship. This is suggested to avoid problems of force onset detection and 150-millisecond windows have been shown to have the best reliability (*Maffiuletti et al., 2016*). Maximum torque was defined as the peak in the filtered force signal. The best performance for both parameters over the three trials was used for analysis. CoV's of around 10% have been reported for RTD and maximum force measurements (*Clark, Cook & Ploutz-Snyder, 2007*).

Participants' endurance capacity was determined with the Six-Minute Walk Test (6MWT). Seniors were instructed to walk as far as possible during a 6-minute period prohibiting running. Participants shuttled between two cones that were placed 30 m apart. Markings were placed every three meters and participants had to stop at the nearest marker

upon the stop signal. CoVs for the 6MWT have been reported between 1.7 and 5.0% (*Kervio, Carre & Ville, 2003*).

Data are provided as means with standard deviations (SD). Linear mixed models were used to analyse the effect of intervention allocation on changes over time. Gender and baseline performance was used as covariate for the analysis. Random intercepts and slopes were used in the model to account for different performance trajectories over time between the subjects. The estimate of the interaction between time and group with 95% confidence intervals is reported together with the corresponding standardized effect sizes (ES). These estimates (beta coefficients) can be interpreted as the difference in change over time between the groups. ES were interpreted as follows: 0 to 0.2: trivial, >0.2: small, >0.6: moderate and >1.2: large (*Hopkins et al., 2009*). All tests were conducted using R (3.5.5) utilizing the packages "lmSupport" (2.9.13), "car" (3.0-2), and "lme4" (1.1).

## RESULTS

Participant flow is provided in Fig. 1. Fifteen participants were allocated to each intervention group. Twelve (AGI) and 15 (TSB) participants finished the intervention as well as the pre- and post-test. The agility group had three dropouts due to loss of contact (1) and personal reasons (2). One person experienced severe dizziness during the agility-training and discontinued the intervention, but attended the post-test and, thus, was included in the analysis. The adverse event happened at the end of the intervention period. Baseline demographic data and outcomes at baseline are shown in Table 2. Participants took part in $90 \pm 8\%$ (AGI) and $91 \pm 7\%$ (TSB) of the sessions.

The agility-training group, on average, improved in several performance parameters (Tables 3 and 4) while declining in the perturbation balance task, trunk flexion, trunk rotation, and trunk extension maximum torque. Notable improvements were found for 6MWT distance (+15.9%), Y-Balance composite score (+4.3%), left and right plantar flexion RTD (+35.8% and +33.2%, respectively), as well as trunk extension RTD (+15.3%).

The traditional strength and balance training group also, on average, improved in most performance aspects except in the perturbation balance task, trunk rotation to the left side, trunk flexion maximum torque and trunk extension RTD. 6MWT distance improved by 12.6%, Y-balance composite score by 4.9%, plantar flexion RTD by 6.8% (left) and 19.7% (right). Dorsal flexion parameters improved in the TSB group between 12.5% and 23.3%.

Regarding between-group differences in change scores, the confidence intervals of the effect sizes implicate a compatibility of the data with trivial to at least medium differences in favour of the agility-training group for the right plantar flexion maximum strength (ES = 0.41, $p = 0.11$) and plantar flexion RTD of both legs (right: ES = 0.43, $p = 0.19$; left: ES = 0.55, $p = 0.11$) as well as trunk extension RTD (ES = 0.35, $p = 0.10$). For the 6MWT, we observed an ES of 0.18 with the confidence interval indicating that a trivial to medium effect in favour of the agility-training group would be compatible with the data. On the other hand, effects in favour of the TSB group that could be between trivial and medium in size were observed for the right dorsal extension maximum strength (ES = 0.32, $p = 0.17$) and the trunk rotation maximum torque to the right (ES = 0.29, $p = 0.06$). For all other
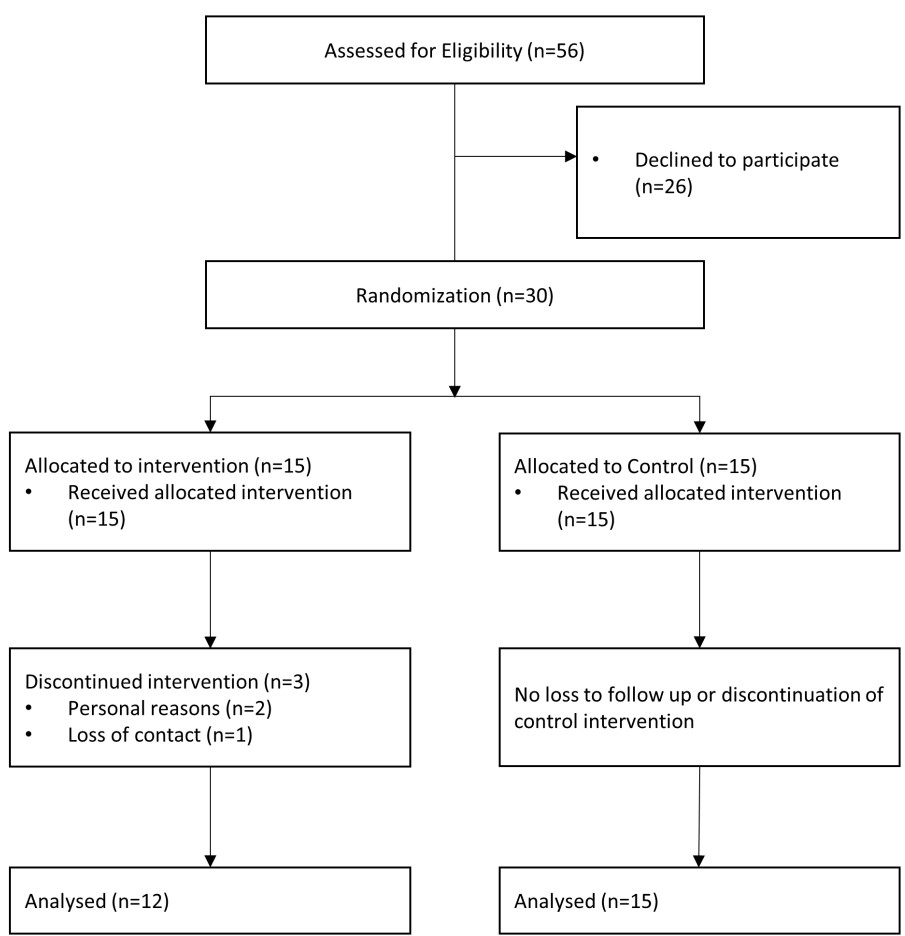

**Figure 1  Flow chart of the study participants.**

**Table 2  Participant characteristics.**

|  | TSB | AGI |
|---|---|---|
| n | 15 | 12 |
| Age (years) | 69.7 (5.4) | 69.3 (5.6) |
| Female | 7 (46%) | 4 (25%) |
| BMI (kg/m$^2$) | 26.4 (3.1) | 26.4 (4.4) |
| Six minute walk (m) | 644 (131) | 706 (230) |
| Compliance (%) | 91 (7) | 90 (8) |

**Notes.**
TSB (traditional strength and balance training), AGI (agility training), Data are mean and SD.

parameters, the confidence intervals of the effect sizes indicate that the data would be compatible with small effects sizes favouring either intervention or being trivial.

Lichtenstein et al. (2020), *PeerJ*, DOI 10.7717/peerj.8781

Peer J

**Table 3 Results of the endurance and balance assessments.** 6-MWT (six minute walk test), CS (composite score), SLEO (30 s single leg eyes open), Perturbation (platform sway path distance during 10 s after platform release). Effect sizes are standardized estimates of the linear mixed effects model for the group factor. Data are mean and SD for pre and post values and mean and 95% confidence intervals for changes and differences.

| | Agility based training | | | Traditional strength and balance training | | | Mixed model analysis | |
|---|---|---|---|---|---|---|---|---|
| | **Pre** | **Post** | **Change** | **Pre** | **Post** | **Change** | **Estimate** | **Effect Size** |
| 6-MWT (m) | 706 (230) | 819 (200) | 113 (74; 151) | 644 (131) | 725 (137) | 81 (36; 156) | 32 (−29; 93) | 0.18 (−0.16; 0.51) |
| Y-Balance CS | 86.7 (12.5) | 90.4 (12.2) | 3.7 (−1; 8.4) | 82.9 (13.9) | 86.9 (14.6) | 4.1 (0.7; 7.5) | −0.4 (−6.0; 5.3) | −0.03 (−0.49; 0.43) |
| SLEO (mm) | 2076 (411) | 2001 (454) | −76 (−217; 66) | 2574 (870) | 2504 (1238) | −70 (−427; 288) | −10 (−427; 415) | −0.01 (−0.58; 0.56) |
| Perturbation (mm) | 448 (177) | 504 (94) | 56 (−18; 131) | 461 (134) | 491 (86) | 29 (−30; 89) | 27 (−68; 122) | 0.08 (−0.20; 0.37) |

Lichtenstein et al. (2020), *PeerJ*, DOI 10.7717/peerj.8781

**Table 4  Results of the strength assessments.** Maximum torque values are in N, rate of torque development (RTD) values in N/150ms. PF (Plantar Flexion), R (right), L (left), DF (dorsiflexion), Rot (trunk rotation), Flex (trunk flexion), Ext (trunk extension). Effect sizes are standardized estimates of the linear mixed effects model for the group factor. Data are mean and SD for pre and post values and mean and 95% confidence intervals for changes and differences.

| | Agility based training | | | Traditional strength and balance training | | | Mixed model analysis | |
|---|---|---|---|---|---|---|---|---|
| | Pre | Post | Change | Pre | Post | Change | Estimate | Effect Size |
| PF Max Torque R | 105.8 (22) | 118.2 (16.9) | 12.4 (−0.7; 25.4) | 90.6 (31.0) | 91.3 (26.3) | 0.8 (−5.8; 7.4) | 11.5 (−2.0; 25.1) | 0.41 (−0.07; 0.89) |
| PF Max Torque L | 99.8 (24.2) | 107.7 (20.0) | 7.9 (−2.3; 18.1) | 94.8 (36.3) | 96.9 (28.8) | 2.2 (−7.5; 11.8) | 5.7 (−8.4; 19.8) | 0.18 (−0.27; 0.64) |
| PF RTD R | 40.3 (18.8) | 53.7 (20.4) | 13.4 (2.8; 24.0) | 28.7 (16.2) | 34.4 (17.3) | 5.7 (0.2; 11.1) | 7.7 (−3.5; 18.9) | 0.43 (−0.19; 1.05) |
| PF RTD L | 37.7 (19.7) | 51.2 (18.9) | 13.5 (2.5; 24.5) | 37.7 (17.5) | 40.3 (18.2) | 2.6 (−3.7; 8.9) | 10.3 (−1.9; 25.5) | 0.55 (−0.10; 1.36) |
| DF Max Torque R | 27.2 (6.5) | 30.6 (9.7) | 3.4 (−0.8; 7.5) | 27.2 (11.3) | 33.5 (11.2) | 6.3 (4.5; 8.2) | −3.0 (−7.1; 1.1) | −0.32 (−0.76; 0.12) |
| DF Max Torque L | 31.7 (9.5) | 34.1 (8.5) | 2.4 (−1.7; 6.5) | 32.3 (11.3) | 36.7 (11.4) | 4.4 (1.7; 7.1) | −2.0 (−6.7; 2.7) | −0.19 (−0.65; 0.26) |
| DF RTD R | 20.0 (7.5) | 21.0 (7.1) | 1.0 (−1.5; 3.5) | 19.2 (5.3) | 21.5 (6.1) | 2.3 (0.5; 4.1) | −1.3 (−4.3; 1.7) | −0.21 (−0.70; 0.27) |
| DF RTD L | 19.1 (4.2) | 21.2 (5.9) | 2.1 (0.1; 4.0) | 19.8 (7.0) | 22.3 (7.3) | 2.5 (0.5; 4.5) | −0.3 (−3.2; 2.6) | −0.05 (−0.54; 0.43) |
| Rot Max Torque R | 87.9 (31.9) | 82.2 (27.7) | −5.8 (−11.4; −0.1) | 80.8 (34.2) | 84.3 (37.2) | 3.5 (−3.3; 10.3) | −9.4 (−18.6; 0.0) | −0.29 (−0.58; 0.00) |
| Rot Max Torque L | 90.0 (32.2) | 88.8 (29.1) | −1.2 (−13.5; 11.2) | 80.9 (27.2) | 76.6 (32.6) | −4.3 (−11.3; 2.7) | 3.1 (−10.4; 16.6) | 0.11 (−0.36; 0.57) |
| Rot RTD R | 49.3 (29.2) | 55.4 (33.3) | 6.1 (−5.3; 17.4) | 37.1 (21.6) | 41.7 (24.5) | 4.6 (−4.1; 13.4) | 2.1 (−12; 16.3) | 0.08 (−0.47; 0.64) |
| Rot RTD L | 52.9 (26.5) | 55.2 (28) | 2.3 (−10.1; 14.8) | 37.7 (18.8) | 34.8 (20.6) | −3.0 (−10.1; 4.2) | 5.8 (−8.3; 18.9) | 0.25 (−0.36; 0.81) |
| Flex Max Torque | 82.5 (27.2) | 76.4 (28.3) | −6.1 (−16.2; 4) | 76.4 (22.3) | 70.7 (23.3) | −5.7 (−12.3; 0.9) | −0.4 (−11.9; 11.1) | −0.02 (−0.49; 0.46) |
| Ext Max Torque | 212.4 (55.9) | 211.7 (64.2) | −0.7 (−17.4; 16) | 167.8 (81.5) | 168.7 (65.5) | 1.0 (−12.6; 14.5) | −1.7 (−23.1; 19.7) | −0.02 (−0.32; 0.27) |
| Flex RTD | 60.3 (31.6) | 59.2 (26.0) | −1.1 (−16.3; 14.2) | 43.8 (22.5) | 45.5 (31.7) | 1.7 (−7.9; 11.3) | −2.0 (−19.1; 15.0) | −0.07 (−0.70; 0.55) |
| Ext RTD | 81.4 (41.6) | 93.8 (49.1) | 12.4 (−1.9; 26.7) | 50.8 (36.9) | 49.7 (28.6) | −1.2 (−10.6; 8.3) | 14.3 (−2.1; 30.7) | 0.35 (−0.05; 0.75) |

## DISCUSSION

In this study, we compared the effect of 8 weeks agility-based training with a traditional strength and balance training program in seniors on measures of neuromuscular and endurance performance. Participants regularly attended the sessions and the findings indicate that the integrative agility-based training approach led to similar adaptations in the short term compared with the traditional strength and balance training approach. In some areas, particularly plantar flexion and trunk extension explosive strength, agility-based training might be superior. Both training programs were well received by the participants, which was indicated by a high compliance to either program.

### Key results

Improvements in maximum strength and rate of torque development were potentially smaller in our study compared to evidence from studies with focused strength-only training in seniors. A systematic review and meta-analysis on short-term strength training interventions showed that in maximum strength and rate of force development increases of 15% to 25% can be expected (*Guizelini et al., 2018*). The initial high fitness level (according to the 6MWT performances; ♂: 712 m, ♀: 613 m, normative data: ♂: 560 m, ♀: 505 m (*Bohannon, 2007*) of the participants in our study and the multimodal training in both groups are the most likely reason for this discrepancy. Yet, changes in rate of torque development of the agility-training group were very similar compared to the mentioned strength-only training programs. The potentially favourable adaptations in trunk extension and plantar flexion RTD in the agility-training group might relate to the frequent use of rapid changes of direction and stop-and-go movements within this intervention that can be considered a form of power training. Power training has been shown to lead to favourable adaptations compared to resistance training for improving functional capacity and postural control in the elderly even at low training volumes (*Guizelini et al., 2018*; *Lopes et al., 2016*). The importance of trunk extension and lower limb power for balance, mobility and fall risk has been discussed (*Bohrer et al., 2019*; *Granacher et al., 2013*). The agility training utilises more explosive movements and therefore could be better suited to improve muscular power compared to traditional resistance training while concurrently being beneficial in other parameters. A resistance training focused on explosive execution of movements might yield similar results with regard to muscular power. The age-associated decline in strength amounts to around 2 to 4% per year in older adults (*Mitchell et al., 2012*). In this regard, the observed strength increases of up to 35.8% in our study after only 8 weeks can be considered relevant, potentially rejuvenating "neuromuscular age" by up to 10 years.

Improvements in balance ability were lacking in this study in both groups. Considering earlier findings of a similar interventions from our lab this was unexpected, especially for the traditional strength and balance training group (*Donath et al., 2016b*). In the past, balance training has been considered task-specific (*Kummel et al., 2016*) and protocols for improving balance are recommended to last 11 to 12 weeks with 90 to 120 min of balance training per week (*Lesinski et al., 2015a*). Underpinning those results, a focused balance training did not result in relevant improvements in similar measures as used in our study

after five weeks (*Ruffieux et al., 2017*). Therefore, the interventions at hand might have been too short and unspecific to elicit changes in balance performance.

Increases larger than 10% were found in both groups in the endurance performance assessment which aligns with previous studies investigating the effect of strength and endurance interventions in a similar timeframe (*Burich et al., 2015*). Minimally clinically important differences for the 6MWT have been reported below 10% (*Kwok et al., 2013*). Therefore, the present changes can be considered meaningful for health. A potentially larger training effect for the agility training group (ES = 0.18, 4.8%) was found, which, however, might be alternatively explained by a higher proficiency of the agility training group in change of direction tasks compared to the control group. At baseline, 23 turns of 180° were performed during the 6MWT and the agility training featured more change of direction tasks. This could have led to an improved economy of locomotion in this task resulting in increased 6MWT performances, irrespective of cardio-circulatory adaptations. Nevertheless, the agility training also employed more walking based exercises compared to the control groups. Therefore, an explanation of the potential group differences could also be in the greater stimulus to the cardio-circulatory system during the agility training. It is unlikely however, that the group difference (4.8%) is of clinically meaningful magnitude.

## Limitations

Some limitations of this investigation have to be mentioned. Results from the initial 6MWT suggest very fit individuals included in this study when compared with normative data (*Bohannon, 2007*). One reason for this are the exclusion criteria. In this pilot trial, we excluded participants with cardiological, neurological and orthopaedic conditions, which are quite prevalent in this age range. We did this as the agility-training is a preventive approach. Therefore, results are not generalizable to all elderly people, but only those who are physically fit and healthy. Adaptations in both study arms might have been greater in less trained individuals, but as this was a pilot trial, where we were also looking into the feasibility of the training program, we decided to only include generally healthy participants. Expansion and adaptation to less fit populations can be considered in the case of positive results. The exercises could be scaled up and down in intensity and duration according to the specific needs of a target population or individual.

The proposed training program includes various exercises that could be considered unsafe for elderly participants as high impact activities are combined with challenging multi-tasks. Apart from one adverse event due to dizziness, no further adverse events were observed. The need for professional supervision seems evident for proper execution. Exercises where the risk of falling might be elevated can be made safer by providing the means to grasp a rail or provide a soft landing surface. In two recent reviews, the safety and efficacy of jumping and plyometric exercises for elderly was pointed out (*Moran, Ramirez-Campillo & Granacher, 2018*; *Vetrovsky et al., 2018*). No sample size estimation was conducted and the study might therefore be underpowered to detect relevant changes. As this was designed as proof-of-concept of the agility-training approach, a convenience sample (two independent training groups) was used. This study serves as a basis for a long-term intervention study and the herein obtained effect sizes in this short-term intervention

seem promising. Some aspects should be considered for long-term implementation. The difficulty of the strength components should also be increased by progressing the intensity utilizing higher weights or speeds. This has not been done in this study due to the short duration of the study.

The lack of objective measurement of training characteristics might be of concern. For a valid comparison of two exercise interventions, usually the training load has to be matched. If not, intervention effects might be alternatively explained by differences in overall training load (*Hecksteden et al., 2018*). For strength, endurance and balance training, dose–response relationships have been investigated suggesting higher load to be related to larger improvements (*Huang et al., 2016*; *Lesinski et al., 2015b*; *Silva et al., 2014*). Establishing the training load during an agility-intervention is rather complex as exercises are performed simultaneously and comparing the dose of a single-component exercise to a multi-component exercise featuring this exercise might be misleading. Nevertheless, heart rate data, rating of perceived exertion and eventually inertial measurement unit data often utilised in team sports should be considered in future studies.

## CONCLUSIONS

We compared the adaptations to two exercise based interventions in older adults and found similar changes in both groups. The agility training might lead to favorable adaptations in explosive power of some muscle groups. The long-term investigation of this integrative multi-modal exercise-training program also with regard to cognitive performance, control of locomotion, muscle architecture and "hard" endpoints like falls or institutionalization should be considered in future research. The agility training approach could be regarded as a time efficient alternative for exercise training in older adults as all relevant aspects of human performance in ageing are trained simultaneously.

## ACKNOWLEDGEMENTS

We would like to acknowledge all participants of both groups and the students conducting the intervention and performing the assessments.

### Funding
The authors received no funding for this work.

### Competing Interests
The authors declare there are no competing interests.

### Author Contributions
- Eric Lichtenstein performed the experiments, analyzed the data, prepared figures and/or tables, authored or reviewed drafts of the paper, and approved the final draft.
- Mareike Morat conceived and designed the experiments, authored or reviewed drafts of the paper, and approved the final draft.

- Ralf Roth performed the experiments, prepared figures and/or tables, and approved the final draft.
- Lars Donath conceived and designed the experiments, analyzed the data, authored or reviewed drafts of the paper, and approved the final draft.
- Oliver Faude conceived and designed the experiments, analyzed the data, prepared figures and/or tables, authored or reviewed drafts of the paper, and approved the final draft.

### Human Ethics

The following information was supplied relating to ethical approvals (i.e., approving body and any reference numbers):

The ethics committee of north-western and central Switzerland (EKNZ) approved the conduction of the study within the facilities of the Department of Sport, Exercise and Health (2017-07940).

### Data Availability

The data is available in the Supplemental Files.

### Supplemental Information

Supplemental information for this article can be found online at http://dx.doi.org/10.7717/peerj.8781#supplemental-information.

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
