# Peer review of "Agility-based exercise training compared to traditional strength and balance training in older adults: a pilot randomized trial"

_PeerJ, doi:10.7717/peerj.8781_

## Round 0.1 · original submission · Minor Revisions

The two reviewers and I congratulate the authors on a well-designed and written study. Please attend to the constructive criticisms of the two reviewers when you look to resubmit the revised version of this manuscript.

·

Basic reporting

This article is well written, uses relevant and current references and the results reflect the hypotheses.
I suggest two minor language changes:
1: Line 86 - masked assessors is now the preferred term - rather than blinded.
2: Please consider an alternative word to "Corroboratory" on line 292.

Experimental design

The research represents a clinically relevant research question which aligns with the Aims and Scope of the journal.
The investigations is well though out and well conducted.
The methods section is well described.
1. Please consider also outlining, in the "General Design" section, the distribution of the training sessions to allow for replication. For example, 24 training sessions (3 sessions per week for eight weeks). Each session was 50 minutes in duration. (LINE 81)
2. It is noted that the Participants were excluded for "cardiologic, neurological and orthopaedic conditions" (LINE 91). Can you please expand on whether this means specific conditions or all conditions. If it means all conditions under these categories it will rule out many older adults and therefore reduce scalability of findings. This should therefore be noted as a limitation in the results section of the paper.
Under the Interventions section please consider describing whether the difficulty was also increased by increasing resistance (not just repetitions) and if not, why not as this is usual practice for strength training programs. (LINE 125)
3. Figure 1: Please include details on the 26 excluded people. How many declined and how many didn't meet the inclusion criteria.

Validity of the findings

The underlying data has been provided and are robust, statistically sound and controlled.
I have the following questions:
1. There were large differences in the baseline measures for 6mwt and SLEO for each group. Can you please comment on whether you think this may have affected your findings.
2. Please describe your cut off score for p value significance for between group differences in change scores. They appear to be 0.1 or 0.2 in some cases and we need to agree on whether that can be determined as significant.
3. The Agility Group had a much bigger change in Perturbation measures, although the CI are wide, can you please comment on whether you think this may be clinically relevant even if not statistically so?
4. Background Conclusion: please provide references that state plantar flexion strength and trunk trunk extension RTD are falls risk factors. Otherwise it may be better to say that :agility- based training seems at least as efficacious as traditional strength and balance training in affecting selected physical performance indicators among community dwelling healthy seniors"
5. I believe that the fact that the agility based training is much more time efficient, while potentially delivering comparable outcomes may be the key finding of this study. The authors have mentioned this but may like to expand on the point. This finding is important as current best practice now recommends up to 3 hours per week to achieve a falls prevention effect (Sherrington et al, 2017). This is very hard to achieve in the real world so this pilot study contributes an important finding to push for the implementation of a larger study to test whether the use of the intervention achieves a falls prevention outcome. Such a finding has the potential to increase uptake and adherence, which would be a very important clinically.

Additional comments

Thank you for conducting this pilot study, and submitting this well written manuscript. You are to be congratulated on designing an intervention that has the potential to reduce the time commitment required by older adults to maintain or improve their physical performance and potentially reduce falls. I have made a few comments for your consideration and would be happy to review again if required.

·

Basic reporting

Basic reporting is clear and professional throughout.

Experimental design

Well designed study but further description is required to replicate.

Validity of the findings

Findings need to be better aligned with expected change data - what is clinically important?

Additional comments

I commend the authors on an interesting an important study assessing the effect of agility training on balance and strength in older adults. I understand that the study was a randomised pilot in nature and only certain inferences can be made with a small sample size, but I believe it could be strengthened further, particularly by including more details about the interventions. Please see my specific comments attached.

---

## Round 0.2 · accepted · Accept

Thanks for addressing the comments of the reviewers. I am now happy to recommend this paper be accepted for publication.

·

Basic reporting

No comment

Experimental design

No comment

Validity of the findings

No comment

Additional comments

Thank you for your responses to my previous feedback, I believe this paper is now ready to be accepted for publication and provides a valuable contribution to falls prevention research.

·

Basic reporting

Basic reporting is clear and professional throughout.

Experimental design

Well designed study with appropriate detail in revised version.

Validity of the findings

Appropriate and clinical effects discussed.

Additional comments

Well done on a well conducted novel training program. Thank you for addressing my concerns/questions - you have amended appropriately and made the paper easier to follow/replicate.